# Arabinofuranosyl Thymine Derivatives—Potential Candidates against Cowpox Virus: A Computational Screening Study

**DOI:** 10.3390/ijms24021751

**Published:** 2023-01-16

**Authors:** Ahlam Haj Hasan, Gagan Preet, Bruce Forbes Milne, Rainer Ebel, Marcel Jaspars

**Affiliations:** 1Marine Biodiscovery Centre, Department of Chemistry, University of Aberdeen, Aberdeen AB24 3UE, Scotland, UK; 2Department of Medicinal Chemistry and Pharmacognosy, Faculty of Pharmacy, Jordan University of Science and Technology, Irbid 22110, Jordan; 3CFisUC, Department of Physics, University of Coimbra, Rua Larga, 3004-516 Coimbra, Portugal

**Keywords:** cowpox, poxviruses, thymine, arabinofuranosyl thymine, molecular docking, pharmacophore, ADMET, zoonotic, virtual screening

## Abstract

Cowpox is caused by a DNA virus known as the cowpox virus (CPXV) belonging to the *Orthopoxvirus* genus in the family Poxviridae. Cowpox is a zoonotic disease with the broadest host range among the known poxviruses. The natural reservoir hosts of CPXV are wild rodents. Recently, the cases of orthopoxviral infections have been increasing worldwide, and cowpox is considered the most common orthopoxviral infection in Europe. Cowpox is often a self-limiting disease, although cidofovir or anti-vaccinia gammaglobulin can be used in severe and disseminated cases of human cowpox. In this computational study, a molecular docking analysis of thymine- and arabinofuranosyl-thymine-related structures (**1–21**) on two cowpox-encoded proteins was performed with respect to the cidofovir standard and a 3D ligand-based pharmacophore model was generated. Three chemical structures (PubChem IDs: 123370001, 154137224, and 90413364) were identified as potential candidates for anti-cowpox agents. Further studies combining in vitro and in silico molecular dynamics simulations to test the stability of these promising compounds could effectively improve the future design of cowpox virus inhibitors, as molecular docking studies are not sufficient to consider a ligand a potential drug.

## 1. Introduction

Cowpox is caused by a DNA virus known as the cowpox virus (CPXV), which belongs to the *Orthopoxvirus* genus in the family Poxviridae. This family includes several species such as Mpox (formerly named monkeypox) (MPXV), camelpox (CMLV), variola (VARV), and vaccinia (VACV) [1]. Cowpox is endemic in Eurasia, mainly in Europe [2]. Over the last 20 years, the number of animal cases of cowpox has been increasing in Europe, and nowadays, cowpox is considered the most common orthopoxviral infection in Europe [3]. Cowpox is a zoonotic disease with the broadest host range among known poxviruses. Natural reservoir hosts of CPXV are wild rodents such as vole and mouse species [4]. Additionally, many species can be naturally infected by CPXV, such as cats, dogs, horses, zoo animals, and humans [5]. Cowpox virus (CPXV) can be transmitted to humans primarily through direct contact with infected animals, mainly infected pet rats [6] and cats [7]. Human-to-human transmission has not been reported [8]. 

The first zoonotic case of cowpox was reported in 1985, and it was diagnosed in the Netherlands, where the CPXV was transmitted to a woman via direct contact with an infected cat [1]. Cowpox is often a self-limiting disease, manifested by localised vesicular-pustular lesions on the hands, face, and neck that usually progress to crust over 2–4 weeks. Fever, malaise, fatigue, and headache often occur after lesions first appear [8,9]. Cowpox can be fatal in high-risk patients, such as immunocompromised patients and patients with underlying atopic skin conditions like eczema [4,10]. In 1980, after the successful eradication of smallpox worldwide, the vaccination against the Variola virus (VARV), which causes smallpox, was massively decreased. This lack of vaccination led to re-emerging other orthopoxviral infections (OPV), such as Mpox (formerly named Monkeypox) and cowpox, which cause health threats, especially for unvaccinated people [10,11]. Recently, cases of orthopoxviral infections have been increasing worldwide, such as cowpox in Europe and Asia, Mpox in Africa, and vaccinia in Brazil and India [12]. In 2019, an atypical case of cowpox was diagnosed in France for a previously smallpox-vaccinated patient. The patient had a mild injury with a metallic object, but the wound failed to heal and became worse with severe symptoms, which were then diagnosed to be due to cowpox as the diagnostic analysis showed atypical novel orthopox virus related to cowpox cladeE3 [10,12]. 

In general, the diagnosis of orthopoxviruses (OPV), including cowpox, can be performed using electron microscopy to detect viral particles and quantitative real-time PCR (qPCR) to detect viral DNA, as well as ELISA; these techniques are expensive and require qualified laboratories [13]. Easy-to-perform and inexpensive techniques can also be used for the diagnosis of OPV, such as the ABICAP (antibody immunocolumn for analytical processes) immunofiltration [13] and dot immunoassay [14] techniques. The treatment of human cowpox is usually supportive, as this disease is self-limiting. Cidofovir or anti-vaccinia gammaglobulin can be used in severe and disseminated cases of human cowpox [15,16]. Cidofovir can be used as a treatment and as a short-term prophylactic therapy for immunocompromised people, especially during the cowpox outbreak [16,17]. 

The computational approach has been broadly used in drug discovery for its advantages in saving time and costs during research, providing a well-established tool that can be applied for many tasks [18,19]. Pharmacophore modelling is one of the most important computational tools, and it has been used extensively in virtual screening, lead optimisation, activity profiling, target identification, and the de novo design of ligands [20]. Pharmacophore modelling is usually accompanied by other computational tools such as molecular docking [21]. Molecular docking is a computational tool that provides a better understanding of the interactions between compounds and their target proteins at the atomic level [22]. These computational tools are very efficient, as they facilitate the screening of many compounds to identify the most promising compounds that can be extensively studied in vitro and in vivo [21]. The in silico ADMET (absorption, distribution, metabolism, excretion, and toxicity) method is another computational tool that can predict the pharmacokinetics and toxicity of compounds [23]. Lipinski’s rule is particularly important, as it evaluates the solubility and permeability of compounds using the following factors: molecular weights less than or equal 500; log P values less than or equal 5; H-bond acceptors less than or equal 10; H-Bond donors less than or equal 5. Passing Lipinski’s rule helps the researchers to focus on compounds which have a good probability of being active orally in human diseases [24].

Pyrimidine and its derivatives were previously reported to have antiviral activity [25,26]. Nucleosides are organic compounds composed of a nucleobase, either pyrimidine or purine, attached to a five-carbon sugar. In general, nucleoside analogues are clinically used as antiviral, antibacterial, and anticancer agents, and they are safe and well tolerated, such as cidofovir, lamivudine, and acyclovir [27,28]. Thymine is a pyrimidine nucleobase found in the DNA, and it is also known as 5-methyluracil [29,30]. A thymine derivative, arabinofuranosyl thymine (Ara-T) (**1**), showed antiviral activity against the cowpox virus, with an EC_50_ of 1.0 μM and IC_50_ of 95 μM, and cidofovir was used as a control compound in this study, which showed anti-cowpox activity, with an EC_50_ of 1.1 μM and IC_50_ of 180 μM [31]. Based on the inhibition shown by a thymine derivative against CPXV in previous research [31], we conducted a computational study of twenty-one related chemical structures (Figure 1) docked against two protein structures. The molecular docking approach was used to predict the binding strengths between the tested structures and two essential proteins of the cowpox virus [32,33]. Based on the results, pharmacophore modelling was performed to create a ligand-based pharmacophore model.

## 2. Results

### 2.1. Molecular Docking

A molecular docking study was performed to predict the relative binding affinities of 21 thymine-related structures (Appendix A). Cidofovir was used as the standard, as this compound is known to inhibit the cowpox virus. Two proteins were used for docking in this study. The first protein was PDB: 4HKJ [32,34], a cowpox-encoded CPXV203 protein (a compact beta-sandwich-structured protein) that binds directly to different major histocompatibility complex class I (MHCI) proteins to form stable complexes and prevent the expression of the MHCI antigen on the surface of the infected cell, thereby preventing T-cell detection and the killing of cowpox-infected cells, as MHCI antigen processing and presentation is an important defence mechanism for the detection and killing of virally infected cells. The second one was PDB: 4PDC [33,35], a cowpox-encoded orthopoxvirus MHC class I-like-protein (OMCP) that binds directly to NKG2D, preventing the activation of NKG2D-bearing lymphocytes from killing the virally infected cell (Figure 2). The binding energy between the structures with cowpox-encoded proteins was analysed and compared to the standard cidofovir. In addition, the interacting residues of protein–structure complexes and the bonding interactions involved were investigated (Figure 3 and Figure 4). Thus, the most promising structures were determined to be structures (**3**), (**5**), and (**11**), depending on the binding energy score, the interacting residues, and the binding interactions with cowpox target proteins (Table 1) (Figure 3 and Figure 4). The convergence method was used in the molecular docking analysis.

Positions (1′) and (4′) of arabinofuranosyl thymine derivatives play a vital role. Therefore, changes in the substituents at position (1′) or position (4′) lead to a change in the affinity with the binding site of cowpox target proteins. Three chemical structures (**3**), (**5**), and (**11**), showed higher binding potential than other derivatives, as these structures have substituents containing a fluorine atom either at position (1′) or position (4′). In contrast, the structures that lack an arabinofuranose ring, such as thymine and its derivatives (**12–21**), showed a lower affinity with the binding site of the target proteins.

The interacting residues of cowpox CPXV203 protein and the types of interactions with the standard cidofovir and the most promising structures (**3**), (**5**), and (**11**) are summarised in Figure 3 and Table 2. Cidofovir showed hydrogen bonding and hydrophobic interactions with cowpox CPXV203 protein residues, including Lys15 with distances of 2.81 and 3.16, Ser148 with a distance of 2.97, and Thr163 with a distance of 3.00 Å. Meanwhile, the Arg155, Phe149, Thr158, and Tyr161 residues have only hydrophobic interactions with cidofovir.

Structure (**3**) showed hydrogen bonding and hydrophobic interactions with two residues of cowpox CPXV203 protein: Ser148 with a distance of 2.74 and Thr158 with distances of 2.76 and 2.83 Å. Meanwhile, Arg155, Phe149, and Thr163 were only involved in hydrophobic interactions. 

Structure (**5**) showed hydrogen bonding and hydrophobic interactions with five residues of cowpox CPXV203 protein: Arg155 with a distance of 3.62; Lys17 with a distance of 3.22; Thr158 with distances of 2.82, 3.15, and 3.19; Thr163 with distances of 2.76 and 3.06; and Tyr161 with distances 2.80 and 2.89 Å. Meanwhile, Asn146 was involved only in hydrogen bonding, with a distance of 3.27 Å, and three residues were only involved in hydrophobic interactions, namely Asp164, Glu 162, and Ser148.

Structure (**11**) showed hydrogen bonding and hydrophobic interactions with three residues of cowpox CPXV203 protein: Thr158 with distances of 2.97 and 3.25, Thr163 with a distance of 2.84, and Tyr161 with a distance of 2.70 Å. Meanwhile, four residues were only involved in hydrophobic interactions, namely Arg155, Glue154, Phe149, and Ser148.

The interacting residues of cowpox OMCP protein and the types of interactions with the standard cidofovir and the most promising structures (**3**), (**5**), and (**11**) are summarised in Figure 4 and Table 3. The standard cidofovir showed hydrogen bonding and hydrophobic interactions with three residues of cowpox OMCP protein: Gly34 with a distance of 2.91; Ser18 with distances of 2.83 and 2.96; and Thr16 with distances of 2.91, 3.08, and 3.18 Å. In contrast, Phe65 interacted only with hydrogen bonding, with a distance of 3.13 Å. In addition, five residues were only involved in hydrophobic interactions with cidofovir, namely Arg67, His17, Leu9, Phe62, and Ser63.

Structure (**3**) showed hydrogen bonding and hydrophobic interactions with two residues of cowpox OMCP protein: Arg67 with distances of 2.80, 3.05, 3.12, and 3.14; and Thr70 with distances of 2.88, 2.97, and 3.08 Å. In contrast, Glu75 was only involved in hydrogen bonding, with a distance of 3.11 Å. In addition, five residues had only hydrophobic interactions, namely Gly13, Ile11, Pro69, Thr72, and Val68.

Structure (**5**) showed hydrogen bonding and hydrophobic interactions with four residues of cowpox OMCP protein: Phe62 with a distance of 2.83, Phe65 with a distance of 2.93, Ser18 with a distance of 3.01, and Thr16 with distances of 2.96 and 3.16 Å. Meanwhile, Arg67, Gly34, Leu9, and Ser63 residues showed only hydrophobic interactions with structure (**5**). 

Structure (**11**) showed hydrogen bonding and hydrophobic interactions with three residues of cowpox OMCP protein: Arg67 with a distance of 2.80, Gly34 with distances of 3.04 and 3.05, and Ser59 with a distance of 2.97 Å. Meanwhile, five residues were only involved in hydrophobic interactions, namely Leu9, Phe62, Ser18, Ser63, and Thr16.

### 2.2. Ligand-Based Pharmacophore Model and ADMET Evaluation

A pharmacophore model was created using low-energy conformers of structures (**3**), (**5**), and (**11**). Four main pharmacophoric features were identified: hydrogen bond acceptors (HBAs), hydrogen bond donors (HBDs), hydrophobic features (H), and aromatic ring (AR). Each structure showed its own 2D and 3D pharmacophoric features, which are shown in Figure 5. The alignment of 3D pharmacophoric features of structures (**3**), (**5**), and (**11**) generated a pharmacophore model with common essential features, which included four HBDs, eight HBAs, two Hs, and one AR. (Figure 6). This pharmacophore model was generated with a score of 0.8320 (scale of 0 to 1) (Appendix A).

The ADMET profiles of the promising structures (**3**), (**5**), and (**11**) were analysed and revealed that these structures passed Lipinski’s rule, which indicates that these structures have chemical and physical properties make them more likely to be given orally to humans.

## 3. Material and Methods

### 3.1. Source of Chemical Structures

The arabinofuranosyl-thymine-related structures and thymine-related structures were taken from the PubChem database [36] to identify potential candidates for anti-cowpox agents for further study (Appendix A). The IUPAC names of these structures are as follows: (**1**) 1-[(2R,3S,4S,5R)-3,4-dihydroxy-5-(hydroxymethyl)oxolan-2-yl]-5-methylpyrimidine-2,4-dione; (**2**) 1-[(2R,3R,4R,5R)-4-hydroxy-5-(hydroxymethyl)-3-(2-methoxyethoxy)oxolan-2-yl]-5-methylpyrimidine-2,4-dione; (**3**) 1-[(2R,3R,4S,5S)-5-[fluoro(hydroxy)methyl]-3,4-dihydroxyoxolan-2-yl]-5-methylpyrimidine-2,4-dione; (**4**) 3-[(3R,4R,5R)-3,4-dihydroxy-5-(hydroxymethyl)oxolan-2-yl]-5-methyl-1H-pyrimidine-2,4-dione; (**5**) 1-[(2R,3R,4S,5S)-5-fluoro-3,4-dihydroxy-5-(hydroxymethyl)oxolan-2-yl]-5-methylpyrimidine-2,4-dione; (**6**) 1-[(2R,3R,4R,5R)-4-hydroxy-5-(hydroxymethyl)-3-methoxyoxolan-2-yl]-5-methylpyrimidine-2,4-dione; (**7**) 1-[(2R,3R,4R,5R)-3-fluoro-3,4-dihydroxy-5-(hydroxymethyl)oxolan-2-yl]-5-methylpyrimidine-2,4-dione; (**8**) 1-[(2R,3S,5S)-3-hydroxy-5-(hydroxymethyl)oxolan-2-yl]-5-methylpyrimidine-2,4-dione; (**9**) 1-[(2R,3R,4R,5R)-3-[2-(dimethylamino)ethoxy]-4-hydroxy-5-(hydroxymethyl)oxolan-2-yl]-5-methylpyrimidine-2,4-dione; (**10**) [(3R,4S,5S)-4-hydroxy-5-(hydroxymethyl)-2-(5-methyl-2,4-dioxopyrimidin-1-yl)oxolan-3-yl] hypofluorite; (**11**) 1-[(3R,4R,5S)-2-fluoro-3,4-dihydroxy-5-(hydroxymethyl)oxolan-2-yl]-5-methylpyrimidine-2,4-dione; (**12**) 5-methyl-1H-pyrimidine-2,4-dione; (**13**) 1-(2-hydroxyethyl)-5-methylpyrimidine-2,4-dione; (**14**) 5-fluoro-1-methylpyrimidine-2,4-dione; (**15**) 1-acetyl-5-methylpyrimidine-2,4-dione; (**16**) 5-fluoro-3-methyl-1H-pyrimidine-2,4-dione; (17) 1,3,5-trimethylpyrimidine-2,4-dione; (**18**) 5-fluoro-1H-pyrimidine-2,4-dione; (19) 5-fluoro-1H-pyrimidine-2,4-dione; (**20**) 5-iodo-1H-pyrimidine-2,4-dione; (**21**) 1-(2-hydroxyethenyl)-5-methylpyrimidine-2,4-dione.

### 3.2. Molecular Docking

The molecular docking method was performed using a similar computational approach [37]. A molecular docking study was conducted using the AutoDock Vina software v.1.2.0. [38]. The 3D crystal structures of cowpox-encoded CPXV203 protein (PDB ID: 4HKJ) and OMCP protein (PDB ID: 4PDC) were retrieved from the Protein Data Bank (PDB) database [39]. The box center and dimensions around the interaction site for PDB ID: 4HKJ were 16.2, 7.9, −73.4, and 38.1 × 58.0 × 46.3 Å. For PDB ID: 4PDC these were −0.9, 26.4, 0.4, and 50.7 × 34.7 × 46.2 Å. All dimensions are in Angstrom units. In this study, we used the software LigandScout 4.4.8 (Inte:Ligand) Advanced Software [40] for the protein binding site predictions. This software identifies the assumed binding sites by making a grid around them and then calculates the buriedness value of each grid point on the surface. The final pocket grid has many clusters of grid points close to each other. The isosurface shows empty space in order to make the pocket. The DockPrep tool was used to prepare all structures using the default parameters in Chimera v.1.16 [41]. The following settings were used in the docking study. The binding mode number was 10, the exhaustiveness level was 32, and the maximum difference in energy was 3 Kcal/mol. A docking study was also performed using the same settings mentioned at different exhaustiveness levels of 8, 16, 24, and 32 to check the convergence of this parameter. Chimera v.1.16 [41], LigPlot+ v.2.2. [42], and Samson by OneAngstrom, 2022 [43], were used for the visualisation and calculation of protein–ligand interactions and for the visualisation of 2D ligand–protein complexes.

### 3.3. Ligand-Based Pharmacophore Generation and ADMET Study

LigandScout v.4.4.8 Inte:Ligand Expert [40] was used to construct a 3D ligand-based pharmacophore model. The following default settings for conformation generation were used. The maximum conformation number for each structure was 50, the timeout was 600 s, the RMS threshold was 0.8, the energy window for conformer selection was set to 20, the maximum pool size of conformers was 4000, the maximum fragment build time was 30, the slave memory was set to −1.0, and the number of slave processes was set to 2. To create the ligand-based pharmacophore, the espresso algorithm was used with the default values to start the generation. Ten ligand-based pharmacophore models were made, and the one with the highest score was chosen (Appendix A).

The in silico ADMET profile for the promising structures was evaluated using the ADMET LAB 2.0 software [44] to predict the absorption and permeability properties. The SMILES strings of structures (**3**), (**5**), and (**11**) were fed to the ADMETLAB 2.0 web server to generate their ADMET profiles.

## 4. Conclusions

In this computational screening study, based on the inhibition shown by arabinofuranosyl thymine against CPXV by the Smee research group and according to the molecular docking and pharmacophore analyses, we report that the chemical structures (**3**), (**5**), and (**11**) are potential candidates for anti-cowpox activity. These structures showed good interaction potential with the binding sites of the cowpox-encoded proteins mentioned in this study. A ligand-based pharmacophore model was generated to support the development of anti-cowpox therapeutics that could be used either individually or in combination with another antiviral agent to treat cowpox. Zoonotic diseases are a pressing problem affecting human health worldwide. Recently, global cases of orthopoxviral infections have been increasing, such as cowpox in Europe and Asia; thus, discovering new antiviral agents is essential.

## Figures and Tables

**Figure 1 ijms-24-01751-f001:**
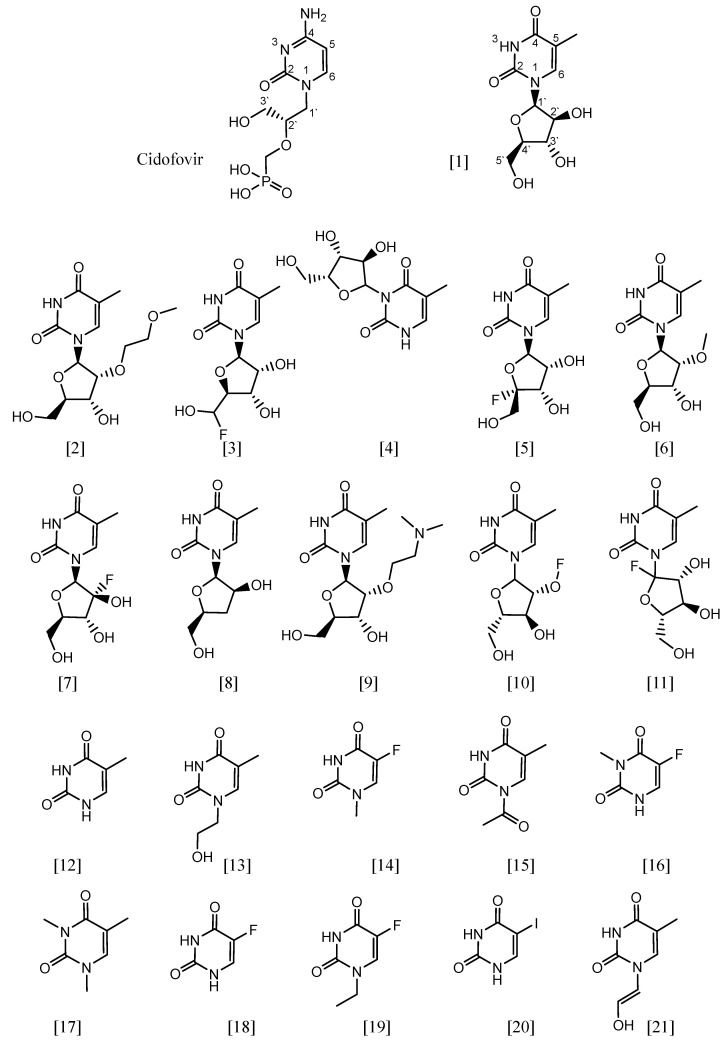
Chemical structures (**1**–**21**) were used in the study and taken from the PubChem database using the Tanimoto score of 97%.

**Figure 2 ijms-24-01751-f002:**
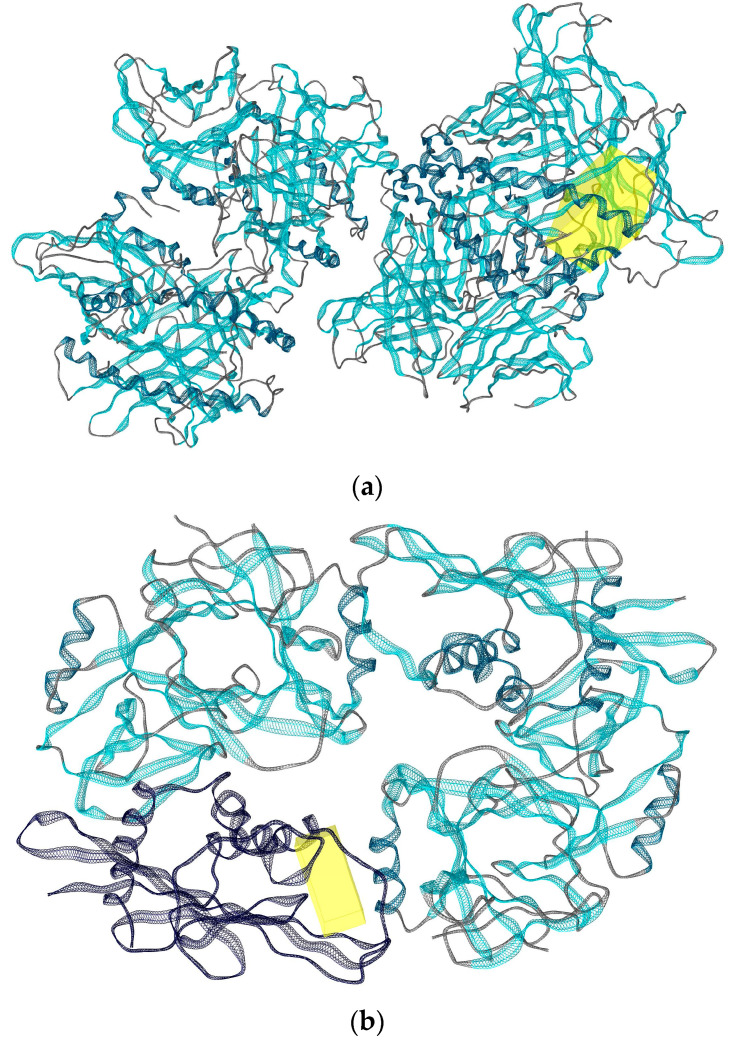
Two cowpox proteins were used in this study: (**a**) the binding site (yellow) of cowpox CPXV203 protein; (**b**) the binding site (yellow) of cowpox OMCP protein.

**Figure 3 ijms-24-01751-f003:**
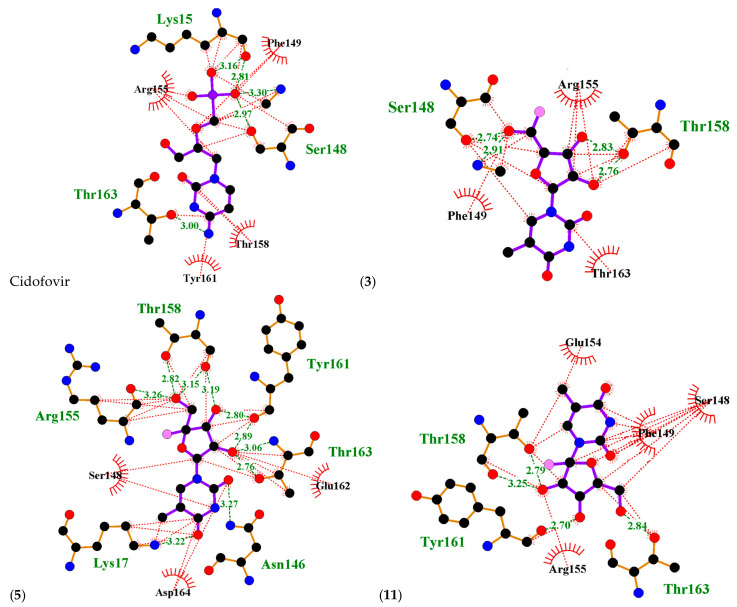
Ligplots showing the interacting residues of cidofovir and arabinofuranosyl-thymine-related structures (**3**), (**5**), and (**11**) with the cowpox CPXV203 protein. Purple lines, thymine structures ligand bonds; orange lines, non-ligand bonds; green dotted lines, hydrogen bonds labelled with distances in Å; red dotted lines, hydrophobic interactions; red circles, oxygen atoms; blue circles, nitrogen atoms; black circles, carbon atoms; pink circle, fluorine atoms; radial lines, non-ligand residues involved in hydrophobic contact(s).

**Figure 4 ijms-24-01751-f004:**
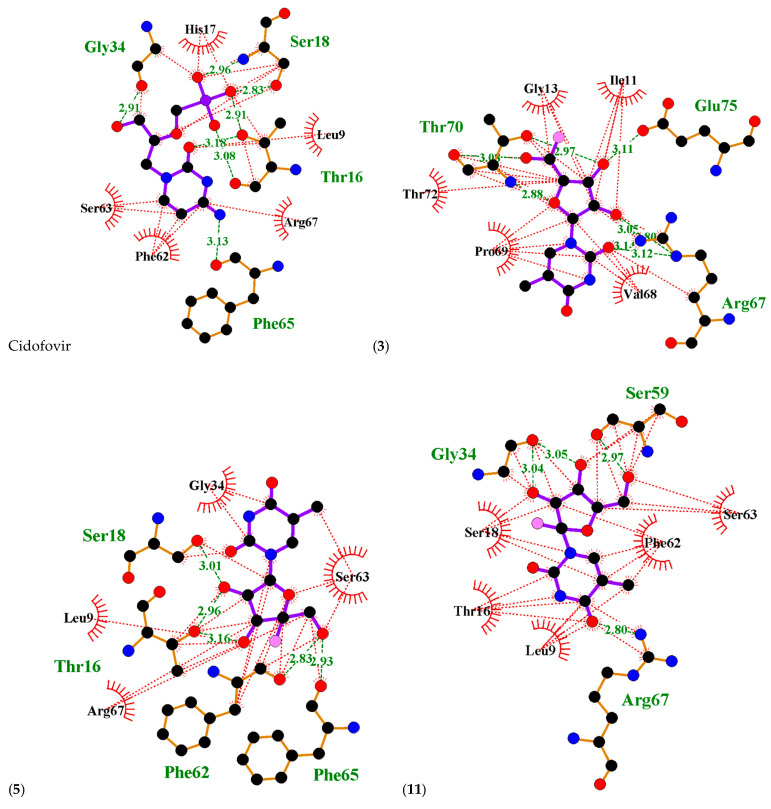
Ligplots showing the interacting residues of cidofovir and arabinofuranosyl-thymine-related structures (**3**), (**5**), and (**11**) with the cowpox OMCP protein. Purple lines, arabinofuranosyl-thymine-related structures ligand bonds; orange lines, non-ligand bonds; green dotted lines, hydrogen bonds labelled with distances in Å; red dotted lines, hydrophobic interactions; red circles, oxygen atoms; blue circles, nitrogen atoms; black circles, carbon atoms; pink circle, fluorine atoms; radial lines, non-ligand residues involved in hydrophobic contact(s).

**Figure 5 ijms-24-01751-f005:**
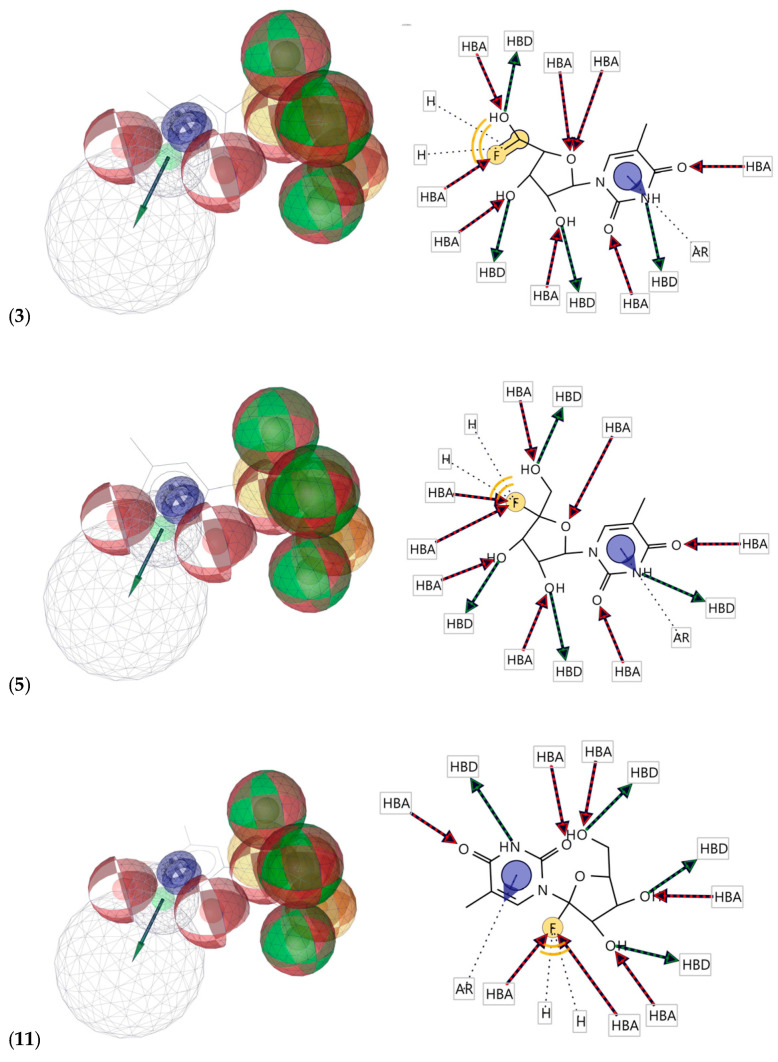
The 2D and 3D pharmacophoric features of structures (**3**), (**5**), and (**11**). Red, HBAs; green, HBDs; purple, AR; yellow, H.

**Figure 6 ijms-24-01751-f006:**
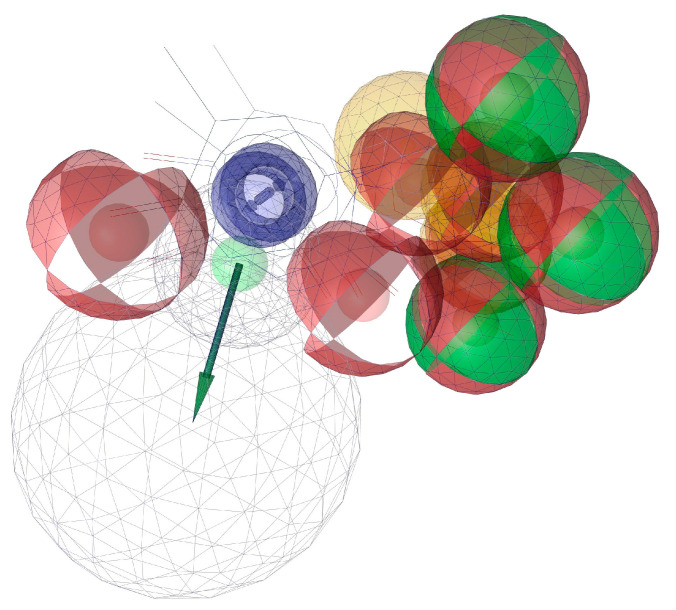
A 3D ligand-based pharmacophore model. Red, HBAs; green, HBDs; purple, AR; yellow, H.

**Table 1 ijms-24-01751-t001:** The molecular docking analysis of twenty-one structures on two different cowpox proteins compared to cidofovir as the standard (shown in blue) and the best lead structures (shown in green).

Compounds (PubChem ID)	Docking Score (-) (Kcal/mol)	Docking Score (-) (Kcal/mol)
	PDB ID: 4HKJ (CPXV203)	PDB ID: 4PDC (OMCP)
**Cidofovir**	5.9	5.5
(**1**) (65049)	5.6	5.8
(**2**) (11243988)	5.7	5.3
(**3**) (123370001)	6.1	5.7
(**4**) (129802452)	5.7	5.4
(**5**) (154137224)	6.2	5.9
(**6**) (191372)	5.6	5.3
(**7**) (21308117)	6	5.3
(**8**) (451892)	5.5	5.2
(**9**) (59119818)	5.4	5.0
(**10**) (90413362)	5.8	5.9
(**11**) (90413364)	6	5.7
(**12**) (1135)	4.8	4.2
(**13**) (566009)	5.1	4.6
(**14**) (78957)	4.5	4.4
(**15**) (667607)	4.9	4.6
(**16**) (330104)	4.8	4.5
(**17**) (78112)	4.3	4.1
(**18**) (3385)	5.0	4.5
(**19**) (348851)	4.4	4.4
(**20**) (69672)	4.4	4.3
(**21**) (163114065)	5.0	4.6

**Table 2 ijms-24-01751-t002:** Showing the interacting residues of cowpox CPXV203 protein (D Chain) with cidofovir and the best structures.

Residues	Cidofovir	Structure (3)	Structure (5)	Structure (11)
Arg155	x	x	x x	x
Asn146			x	
Asp164			x	
Glu154				x
Glu162			x	
Gly34				
Lys15	x x			
Lys17			x x	
Phe149	x	x		x
Ser148	x x	x x	x	x
Thr158	x	x x	x x	x x
Thr163	x x	x	x x	x x
Tyr161	x		x x	x x

Note: x: hydrogen bonding; x: hydrophobic bonding.

**Table 3 ijms-24-01751-t003:** Showing the interacting residues of cowpox OMCP protein (F chain) with cidofovir and the best structures.

Residues	Cidofovir	Structure (3)	Structure (5)	Structure (11)
Arg67	x	x x	x	x x
Glu75		x		
Gly13		x		
Gly34	x x		x	x x
His17	x			
Ile11		x		
Leu9	x		x	x
Phe62	x		x x	x
Phe65	x		x x	
Pro69		x		
Ser18	x x		x x	x
Ser59				x x
Ser63	x		x	x
Thr16	x x		x x	x
Thr70		x x		
Thr72		x		
Val68		x		

Note: x: hydrogen bonding; x: hydrophobic bonding.

## Data Availability

Not applicable.

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
