# Peer review of "Arabinofuranosyl Thymine Derivatives—Potential Candidates against Cowpox Virus: A Computational Screening Study"

_ijms, 2023, doi:10.3390/ijms24021751_

Round 1

Reviewer 1 Report

Molecular docking analysis of thymine and arabinofuranosyl thymine-related structures on two Cowpox-encoded proteins was done with regard to the Cidofovir standard, and a 3D ligand-based pharmacophore model was constructed in this work. The top three three compounds have been discovered as possible anti-cowpox agents. The study conducted is systematic and interesting. However I have few concerns before accepting this paper:

1.       Add the limitation of the current study in abstract section.

2.       Add name of the top three compounds or Pubchem Identifiers in abstract instead of chemical structures (3), (5), and (11).

3.       Since this is a computational study, write the advantages of molecular docking and pharmacophore modeling in introduction.

4.       In Table 2 and figure 4, remove” Ace50 and Nme150 “, non-standard amino acid residues.

5.       I also recommend the authors if possible to perform molecular dynamics simulations for at least 50 nanoseconds for stability analysis accompanied with binding free energy analysis (Since the paper lack intensive computations).

6.       Add a statement about drug likeness scores (Lipinski’s filters) of top three compounds.

Reviewer 2 Report

In this manuscript, Jaspars and co-workers have presented a computational screening of some arabinofuranosyl thymine derivatives, showing an interest to identify potential candidates against Cowpox Virus. The manuscript is well-organized and fluent. Few minor changes required before publication.

1. Revise keywords as in silico or such words do not convey meaningful information to readers.

2. Use of super and subscript such IC50 needs to be revised.

3. use of units such as Angstrom in discussion after every distance is discouraged. delete and use once after all the numerals.

Round 2

Reviewer 1 Report

My comments are satisfactorily addressed.